# Fluorescent Gold Nanoparticles in Suspension as an Efficient Theranostic Agent for Highly Radio-Resistant Cancer Cells

Sarah Vogel [1,2,3], Alice O'Keefe [1,2,3], Léa Seban [4], Michael Valceski [1,2,3], Elette Engels [1,2], Abass Khochaiche [1,2,3], Carolyn Hollis [1,2,3], Michael Lerch [1,2], Stéphanie Corde [1,2,5], Christophe Massard [4], Komla Oscar Awitor [4,*] and Moeava Tehei [1,2,3,*]

1  Centre for Medical Radiation Physics, Faculty of Engineering and Information Science, University of Wollongong, Wollongong, NSW 2522, Australia
2  Illawarra Health and Medical Research Institute (IHMRI), University of Wollongong, Wollongong, NSW 2522, Australia
3  Molecular Horizons, University of Wollongong, Wollongong, NSW 2522, Australia
4  CNRS/IN2P3, Laboratoire de Physique de Clermont (LPC+), UMR 6533, Université Clermont Auvergne, 5 Avenue Blaise Pascal CS 30086, 63178 Aubiere, France
5  Radiation Oncology Department, Prince of Wales Hospital, Randwick, NSW 2031, Australia
*  Correspondence: komla.awitor@uca.fr (K.O.A.); moeava@uow.edu.au (M.T.)

**Abstract:** Gold nanoparticles are a promising candidate for developing new strategies of therapy against cancer. Due to their high atomic number and relative biocompatibility, they are commonly investigated as radiosensitizers to locally increase the dose of radiotherapy. In order to optimize this radiosensitizing effect, it is necessary to control the positioning of the nanoparticles in the cells. The purpose of this study is to investigate, by means of fluorescent gold nanoparticles in suspension, the dose enhancement on highly radio-resistant cancer cells. These nanoparticles were successfully produced using modern click-chemistry methods, first by attaching a chelating agent Diethylenetriamine pentaacetate benzylamine to L-cysteine, bonding the resulting ligand to a gold core, grafting propargylamine and then utilizing copper-catalyzed azide-alkyne cycloaddition (CuAAC) to fuse AlexaFluor 647 to the ligands. The results of this study prove the success of the reactions to produce a minimally cytotoxic and highly stable nanoparticle suspension that increases the radiosensitivity of gliosarcoma 9L tumor cells, with a 35% increase in cell death using 5 Gy kilovoltage radiation. Their fluorescent functionalization allowed for their simple localization within living cells and detection in vivo post-mortem.

**Keywords:** gold nanoparticles; dose enhancement; radiosensitizer; fluorescence; click chemistry; Infra-Red analysis

## 1. Introduction

The combination of nanoparticles with radiotherapy is a known and promising way to enhance current cancer treatments [1–5]. The local and preferential dose increase in malignant cellular tissues caused by nanoparticles allows for better tumor control whilst minimizing the collateral adverse effects on healthy tissues. This local dose enhancement is achieved through the production of secondary electrons, scattered photons, and reactive oxygen species (ROS). By producing short-range high linear energy transfer (LET) electrons and increasing oxidative stress, nanoparticle interactions induce further double strand breaks (DSB) within cancer cell DNA. Glioblastoma cancer stem cells have been shown to have an increased propensity to repair DNA damage [6]. Thus, generating numerous DSBs is of critical importance when treating high-grade cancers, such as brain gliomas and glioblastomas.

There are some possible destructive effects as a result of the use of nanoparticles. These include oxidative stress and cell apoptosis [7,8] within cells as well structural changes in

the proteins within blood plasma [9]. In general, however, the benefits of the use of nanoparticles for the treatment of cancer far outweigh the potential risk of nanoparticle accumulation, particularly considering the already extensive use of nanoparticles.

Numerous physico-chemical characteristics of nanoparticles affect local dose increases, such as their size, the chemical nature of the core, and surface functionalization. In particular, the size of nanoparticles has a direct effect on their uptake by cells; the smaller the size of the nanoparticle, often the greater uptake and retention by cells [10]. Nanoparticle shape also affects uptake, with spheres being shown to accumulate within cells far greater than rod-shaped nanoparticles [11].

The surface of the nanoparticles can be functionalized using ligands such as chemical markers or therapeutic agents that may be attached. By functionalizing the surface, nanoparticles can more efficiently be distributed throughout the body to reach their target. Polyethylene glycol is a popular ligand which increases circulation time within the body and prevents aggregation [12]. Furthermore, the nanoparticle surface can be further functionalized as a drug delivery mechanism [13].

Additionally, the shape and size of the nanoparticle can change their optical and plasmonic properties and thus permits light-matter manipulation [14]. This allows for metal-enhanced fluorescence (MEF), which is used in biosensory applications. However, there are practical limitations to this type of fluorescence [15], with issues with non-specific absorption, colloidal stability, and in general, only presented as a proof-of-concept. Fluorescence is a simple alternative, which is highly useful in the study of nanoparticles in vitro, as it is a common laboratory methodology. It can also be used in vivo coupled with CT to conveniently locate the nanoparticles compared to the bony anatomy of a patient [16]. It is a highly desirable imaging modality as it does not employ any ionizing radiation whilst remaining accessible and highly sensitive [17].

The use of a radiosensitizing nanoparticle suspension consisting of particles <10 nm that can passively accumulate in a brain tumor mitigates the need for drug therapies. The treatment for gliomas and glioblastomas is typically complicated by the inability of chemotherapeutics to breach the blood-brain barrier. Gold nanoparticles (AuNPs) are widely studied as a radiosensitizing agent as they have a high atomic number (Z = 79), relative biocompatibility, and a passive internalization in tumor tissue [18]. Passive targeting exploits the leaky vasculature and endothelium of solid tumors [19] and is commonly used to allow nanoparticles to accumulate in the cytoplasm and surround the nucleus through an endocytic process [20].

AuNPs within cells enhance the radiation damage to the cell as they produce short-range electrons (<1 μm) through Auger cascades and photoelectrons when irradiated with kilovoltage X-rays. This occurs due to a vacancy in the K, L, or M shell of the atom after the photoelectric absorption and hence leads to de-excitation and the emission of Auger electrons or characteristic X-rays [21,22]. The electrons within the Auger cascades have an average energy of less than 5 keV and are highly effective at delivering high doses to their surroundings, including the nucleus. The incidence of the photoelectric effect is strongly associated with the atomic number Z of the material, making AuNPs strong radiosensitizers. Studies have shown that by including a concentration of 1% by mass of gold (gold: solvent), the absorbed dose is doubled when irradiating with lower keV energies [23]. Due to their high Z nature, AuNPs possess several characteristic absorption edges in the low kilovoltage energy range used commonly in computed tomography (CT). This produces a significant contrast between AuNPs and surrounding soft tissue, making them an effective theranostic (combined therapeutic and diagnostic) agent for image-guided radiotherapy [24].

Suspensions of AuNPs are simple to manipulate to target cancerous cells and enhance local radiation dose most effectively. Here we have developed a novel AuNP suspension based upon 'click' chemistry that aims to attach a common fluorophore, Alexa Fluor 647, to the nanoparticle core. Click chemistry refers to a group of reactions that are rapid, simple to create, easy to purify, and produce a very high yield of product [25,26]. This suspension maintains stability within the cellular environment by using a stable suspension as opposed

to a nano-powder. These facets are highly desirable to be able to produce useful therapeutic agents on a large scale.

This work introduces an AuNP fluorescent suspension that is stable long-term and investigates its contribution to both the localization and radiosensitization of kilovoltage radiotherapy for the treatment of radioresistant tumor cells. This suspension is also shown to be detected in vivo post-mortem, demonstrating its future use in radiation treatment animal studies.

## 2. Materials and Methods

### 2.1. Fluorescent Gold Nanoparticles Materials

L-Cysteine was purchased from Acros Organics (Beijing, China). Sodium carbonate, *N*-(3-dimethylaminopropyl)-*N′*-ethylcarbodiimide, *N*-hydroxysuccinimide, Propargylamine, *N*,*N*-dimethylformamide were all purchased from Sigma (Roedermark, Germany). Diethylenetriamine pentaacetate benzylamine was purchased from Alfa Aesar (Osaka, Japan), and Sodium ascorbate from Alfa Aesar (Karlsruhe, Germany). Hydrogen tetrachloraurate (III) hydrate was purchased from Stem Chemical (Newburyport, MA, USA) and Methanol from Elvetec services (Meyzieu, France). Sodium borohydride was purchased from Acros Organics (Göteborg, Sweden). Hydrochloric acid was purchased from Fisher Chemical (Loughborough, UK). Ethyl ether was purchased from Prolabo (Paris, France), and Sodium hydroxide from VWR chemicals (Fontenay-sous-Bois, France). Dimethylsulfoxide was purchased from Fisher bioreagent (Waltham, MA, USA), and Copper sulfate from Acros Organics (Veneto, Italy). AlexaFluor647 was purchased from Invitrogen (Waltham, MA, USA).

All experiments were performed under atmospheric conditions, and materials were used without further purification.

### 2.1.1. L-Cys-DTPA Ligand Synthesis

L-Cysteine (L-Cy) was first deprotonated using an aqueous $Na_2CO_3$ solution. Diethylenetriamine pentaacetate benzylamine (DTPA-Ba) was then added to this solution and stirred at room temperature for 24 h. The obtained powder was dried, redissolved, and filtered.

### 2.1.2. Synthesis of [Au(L-Cys-DTPA)] Nanoparticles

First, the gold salt is dissolved in a methanol-water mixture under strong agitation for 5 min. The synthesized ligand L-Cys-DTPA is then added to the previous solution. After a few minutes of magnetic stirring, ice-cold acetic acid is then added. After 5 min of stirring, the aqueous solution of $NaBH_4$ (182 mg in 13.2 mL $H_2O$) is introduced into the mixture in portions until the reduction in gold is visualized by the appearance of purple color in the reaction medium. The final mixture is stirred for 1 h at room temperature. After one hour of stirring, a solution of hydrochloric acid is added. The mixture is then stirred for a further 15 min. After allowing the solution to stand for 30 min, the reaction medium is filtered under reduced pressure through a membrane with a porosity of 0.22 μm. The resulting black powder is then washed several times with solutions of hydrochloric acid, distilled water, and ethyl ether. The nanoparticle powder is dried under a vacuum in a desiccator for 24 h. The functionalized AuNPs are then easily re-dispersed in a sodium hydroxide solution with a concentration of 10 g/L.

### 2.1.3. Synthesis of [Au(L-Cys-DTPA)(Propargylamine)] Nanoparticles

An aqueous solution of *N*-(3-dimethylaminopropyl)-*N′*-ethylcarbodiimide (EDC) and *N*-hydroxysuccinimide (NHS) in a 1 to 2 in molar ratio, respectively, was added to a colloidal suspension of functionalized AuNPs at a concentration of 1 g/L. After one hour and 30 min of stirring, the pH was adjusted to 8. The solution of propargylamine (35 μL) in 100 μL DMF 35% *v/v*) was then added to the activated colloidal suspension. This allowed Carboxyl-to-amine crosslinking using the carbodiimide EDC and Sulfo-NHS. The

suspension was then purified by dialysis against deionized water for 48 h, with the water changed every twelve hours. The dialysis membrane has a molecular cut-off of 6–8 kDa.

2.1.4. Alexa Fluor Grafting on [Au(L-Cys-DTPA)(Propargylamine)] Nanoparticles

To an alkyne functionalized gold suspension (25 mL; 1 g/L), 2.5 mL of Alexa Fluor 647 solution (0.5 mg/10 mL DMSO) was added. After a few minutes of stirring, the catalyst was added: (1.25 mL $CuSO_4$ (1 mM) and 1.25 mL sodium ascorbate (25 mM), causing the copper-catalyzed Huisgen 1,3-Dipolar Cycloaddition (click reaction) of alkynes to azides to occur. The counter-dialysate (deionized water) was renewed after 3, 15, and 21 h. Following one night under magnetic stirring, the suspension was purified by dialysis, whose membrane has a molecular cut-off of 6–8 kDa.

*2.2. Transmission Electron Microscopy (TEM) and Energy Dispersive Spectroscopy (EDS)*

Scanning transmission electron microscopy (STEM) was performed using a JEOL JEM-ARM200F 200kV probe (Tokyo, Japan) at the University of Wollongong Institute for Superconducting and Electronic Materials (ISEM, Wollongong, NSW, Australia). High-resolution images were taken to visualize the gold core structure, and Energy Dispersive Spectroscopy (EDS) was taken to qualitatively assess the element distribution of the particles. The sample was left to partially dehydrate over two days to increase the concentration of the material before TEM preparation. The aqueous suspension was deposited dropwise to a copper-coated lacey carbon grid and left to dry overnight before analysis. The images were processed using Gatan Digital Microscope, and EDS maps were produced using Noran System 7 (NSS) software.

*2.3. Fourier-Transform Infrared Spectroscopy (FTIR)*

Fourier-transform infrared spectroscopy (FTIR) was taken to confirm the successful synthesis and cross-linking of the AF647 fluorophore. The spectra of the ligand coating were analyzed by a comparison between the molecular structures present in the samples with and without the cross-linked fluorophore. The data were taken using the Bruker VERTEX 80 infrared spectrometer (Bruker, Germany), within the range of 4000 $cm^{-1}$ to 400 $cm^{-1}$ at a resolution of 0.5 $cm^{-1}$. In each case, the background spectrum was measured before the raw transmittance was recorded, and the solid samples were compressed onto the surface of the reflection unit (quartz). The background spectrum was removed from the raw data, and the new spectra were normalized using the S−S transmittance percentage (%) as a benchmark at 501 $cm^{-1}$.

*2.4. UV-Vis Spectroscopy*

The colloidal suspensions were diluted to a gold concentration of 0.1 g/L. Their pH was then adjusted to the desired value by adding 0.25 M and 0.1 M hydrochloric acid or sodium hydroxide solutions. The absorbance between 400 and 700 nm using a Perkin Elmer Lambda 35 double beam UV-vis spectrophotometer (Perkin Elmer, Singapure, Singapore) was recorded using quartz cuvettes to determine the size of the nanoparticles with respect to a change in pH. Ultra-pure water was taken as a reference.

*2.5. Thermogravimetric Analysis*

The functionalized AuNP suspension was evaporated using a Heidolph rotary evaporator VV-micro (Schwabach, Germany). The dried AuNP powder was used for the analysis. The thermogravimetric analysis is carried out using a Perkin Elmer TGA 4000 thermogravimetric analyzer (Perkin Elmer, Groningen, The Netherlands). The analysis was performed at a heating rate of 5 °C/min under a 40 mL/min nitrogen gas flow.

*2.6. Absorption and Emission Spectra*

A 100 µL sample of the AuNP suspension was placed in one well of a 96-well plate (Greiner Bio-One, Kremsmünster, Austria). The plate was then scanned using the Spectra-

Max Plus 384 Microplate Reader (Molecular Devices, San Jose, CA, USA) by varying the laser wavelength from 450 to 750 nm to detect the absorption by the nanoparticles and the emission of the fluorophore.

### 2.7. Cell Culture

9L rat gliosarcoma (9LGS) cells (obtained from the European Collection of Cell Cultures (ECCC)) were cultured in T75 cm$^2$ flasks containing complete Dulbecco's modified eagle medium (c-DMEM from Gibco (Crawley, WA, Australia), with 10% fetal bovine serum (FBS) and 1% penicillin and streptomycin (PS)) and incubated at 37 °C and 5% ($v/v$) CO$_2$. It is an adherent radioresistant cell line derived from an *N*-nitrosomethylurea-induced tumor. The cell line inherited glial morphology and was tested for mycoplasma contamination prior to use with nanoparticles.

### 2.8. Cytotoxicity Assessment

9L cells were subcultured into T25 cm$^2$ flasks prior to being exposed to the AuNP suspension. The media inside the flasks was removed, and the suspension, diluted to 50 μg/mL [3,5] in c-DMEM, was added to the flask. 24 h post addition, the cells were washed and trypsinized (EDTA 0.05% Gibco) before being seeded into 100 mm petri dishes containing 10 mL of complete DMEM. Each flask was seeded in triplicate sets of differing seedings, including a control flask of 9L cells with no nanoparticle treatment. Following a period of 14–15 doubling times, each plate was washed with Dulbecco's Phosphate buffered saline (DPBS) with Ca$^{2+}$/Mg$^{2+}$ and stained with a solution of crystal violet (1:3 ($v/v$), 2.3% crystal violet stock (Sigma Aldrich, St. Louis, MO, USA): 70% ethanol). The stained colonies containing at least 50 cells were counted for each plate, with the surviving colonies compared to the number of initially seeded cells to determine plating efficiency (PE). For each treatment, the surviving fraction (SF) was calculated as the ratio of the PE of treated cells to the ratio of untreated (control) cells

### 2.9. Confocal Imaging

9L cells were cultured into Ibidi 8 well μ slide for 3 days, and all c-DMEM was removed before fresh c-DMEM containing 50 μg/mL or 100 μg/mL of the AuNP suspension was added, with the slide incubated for a further 24 h. Following this incubation, 2 μg of Hoechst 33342 Pentahydrate (bis-Benzimide) (Sigma Aldrich, St. Louis, MO, USA) was added to each well and incubated for a further 30 min. The cells were then exposed to wavelengths 405 nm (for Hoechst) and 633 nm (for Alexa Fluor 647) using a Leica TCS SP8 Confocal Microscopy system with Leica LasX Application Suite (from Leica Microsystems, Wetzlar, Germany).

### 2.10. Flow Cytometry

Control and AuNP at 50 μg/mL treated 9L cells were separately cultured in T25 flasks and were harvested and transferred to individual 15 mL Falcon tubes. The tubes were spun at room temperature in a Heraeus Multifuge X3R centrifuge (Thermo Fischer, Bremen, Germany) at 500× *g* for 5 min, with the supernatant of media discarded.

Cells were resuspended and washed with 500 μL of DPBS, with an aliquot removed. The cells were stained using Dihydrodichlorouorescein diacetate (Fluorescent probe: dichlorofluorescein (DCF)) (Ex/Em: 492 nm/527 nm) at a concentration of 10 μM to detect intracellular ROS. The cells were stained for 30 min before being washed and resuspended again in DPBS. The samples were exposed to 640 nm excitation (for detection of AF647) using the BD LSR FortessaX-20 Flow Cytometer (BD Biosciences, Franklin Lakes, NJ, United States) with the forward and side scatter plotted and the histogram of fluorophore emission obtained.

The aforementioned aliquot was Imaged using the confocal microscopy technique (and exposed to a 488 nm laser for detection of DCF) as described in Section 2.9.

### 2.11. Irradiation of Cell Culture and Clonogenic Cell Survival Assay

9L Gliosarcoma cells were subcultured in T12.5 cm$^2$ flasks before exposure to 50 µg/mL of the AuNP suspension 24 h prior to irradiation. Flasks containing untreated cells and cells treated with only the gold suspension were also maintained as controls. The flasks were transported to the Prince of Wales Hospital (Sydney, NSW, Australia), where they were irradiated with 2 and 5 Gy orthovoltage radiation at 125 kVp and beam current 20 mA at a dose rate of 2Gy/min using the Nucletron Oldelft Therapax DXT 300 Series 3 Orthovoltage X-ray machine (Nucletron B.V., Veenendall, The Netherlands) and filtration of 3 mm beryllium (Be), 0.1 mm copper (Cu) and 2.5 mm aluminum (Al) (HVL = 6.58 mm Al). The tube voltage of 125 kVp (filter 4) was chosen due to the mass-energy absorption of gold compared to water (Figure 1). Cells were harvested from the untreated and treated flasks and the survival was assessed via clonogenic assay, as described in Section 2.8.

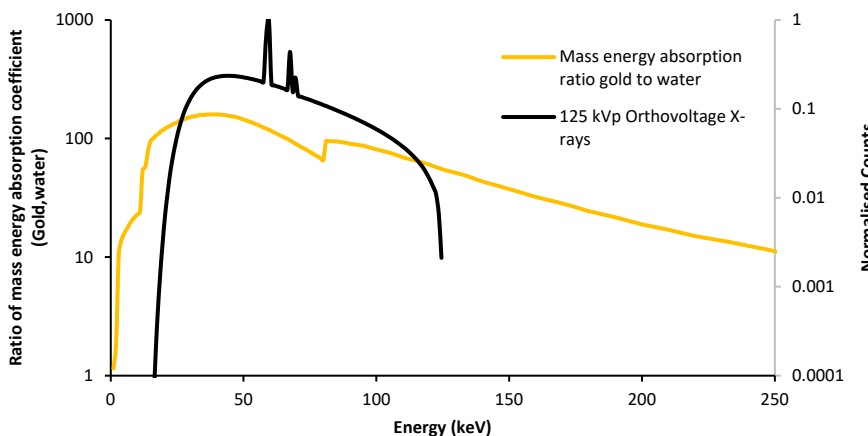

**Figure 1.** The ratio of mass energy absorption coefficient of gold compared to water against the orthovoltage filtration used (filter 4, 125 kVp) [27]. This was chosen as the optimal filter to increase the energy deposited by the interactions and hence increase dose enhancement.

### 2.12. γ-H2AX Analysis

9L Gliosarcoma cells were subcultured into wells of ibidi 8 well micro-slides before exposure to 50 µg/mL of AuNPs in suspension, 24 h prior to irradiation. Wells containing untreated cells and cells treated with only nanoparticles were also maintained. The slides were transported to the Prince of Wales Hospital (Sydney, NSW, Australia), where they were irradiated with 2 and 5 Gy orthovoltage radiation as described above. The cells were incubated at room temp for 20 min (beginning from the start of irradiation) before incubating on ice. Following the period on ice, the wells were rinsed with DPBS and fixed with ice-cold 100% methanol. The wells were washed post-fixation and then treated with a blocking solution of 3% Bovine serum albumin (BSA) in PBS. Cells were then exposed to mouse anti-γH2AX antibody for 2 h. The primary antibody was then washed off before being treated with the secondary goat anti-mouse antibody conjugated with Alexa-488 for 1 h. The chambers were rinsed and exposed to a 488 nm laser and bright field using the Leica TCS SP8 Confocal Microscope (Leica Microsystems, Wetzlar, Germany). Images were taken with a 63x oil objective in a 50-image stack across the depth of the cells. Foci were counted using ImageJ [28], and an average number and intensity of foci per cell were found for each treatment. This number of the average intensity of foci per cell was compared between control and nanoparticle treatments to determine an enhancement ratio.

### 2.13. Injection into Rodent Postmortem

The AuNP suspension was diluted to 50 µg/mL, and 0.5 mL was injected at an approximate depth of 5 mm into the flank of a rat cadaver. The cadaver was then imaged using the IVIS SpectrumCT In Vivo Imaging System (Perkin Elmer, Baesweiler, Germany) at the University of Wollongong, targeting the AF647 excitation.

## 3. Results

### 3.1. Study of the Stability of Colloidal Suspensions as a Function of pH

The phenomenon of plasmon resonance associated with the collective oscillation of electrons in the conduction band causes AuNPs to strongly absorb visible light. The general appearance of the spectra indicates that the suspensions are made up of nanoparticles of a few nanometers in diameter, and therefore the aggregation is low regardless of the pH value (as evidenced in Figure 2). The chemical surface functionalization of the nanoparticles may protect them nanoparticles from aggregation.

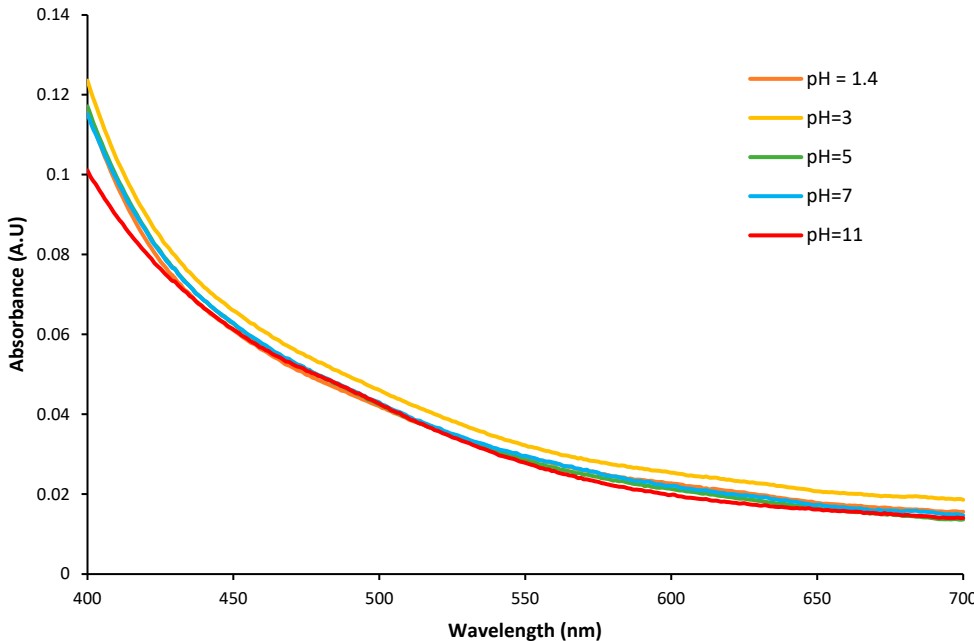

**Figure 2.** UV-visible spectra of colloidal suspensions of AuNPs (at 0.1 g.L$^{-1}$) at different pH.

### 3.2. Thermogravimetric Analysis

Figure 3 shows the evolution of the mass loss of dried AuNPs as a function of temperature. A continuous mass loss is observed up to 700 °C. The observed behavior is most likely a consequence of the surface desorption of the surface chemical functionalization composed of various organic constituents. The figure clearly shows a total weight loss of 63%, with the weight loss (40%) up to approximately 400 °C attributed mainly to water molecules. Dehydration of adsorbed water by the AuNPs powder set in at approximately 120 °C. The water of constitution to the AuNPs powder is eliminated up approximatively to 400 °C, at three times (about 220 °C, 285 °C, and 400 °C). We hypothesize that this is due to the presence of porosities within the nanopowder. The second weight loss is attributed to the functionalized compound, with the remaining mass being gold nanopowder. Based on the initial mass used (14 mg), the approximate hydration of the functionalized gold nanopowder is 7.5%.

### 3.3. Transmission Electron Microscopy (TEM) and Energy Dispersive Spectroscopy (EDS)

The STEM imaging in Figure 4 confirms the successful synthesis of a core-ligand structure, showing a diffuse matrix of organic polymers surrounding gold cores with a diameter between 1–2 nanometers. The localized Au atoms produce weak fringes and are evident as bright clusters, where the strong signal is due to gold's high atomic mass. EDS mapping (Figure 4) shows the distribution of the polymer compounds around the Au core, with an enhanced signal contribution from the carbon support grid. EDS mapping was used qualitatively to visualize the distribution of elements, and while the gold atoms produce a relatively strong signal, the composition of carbon, nitrogen, oxygen, and Sulphur cannot

be accurately quantified due to the elements' low atomic numbers. The elemental mapping of the suspension can be found in Supplementary Material, Figure S1.

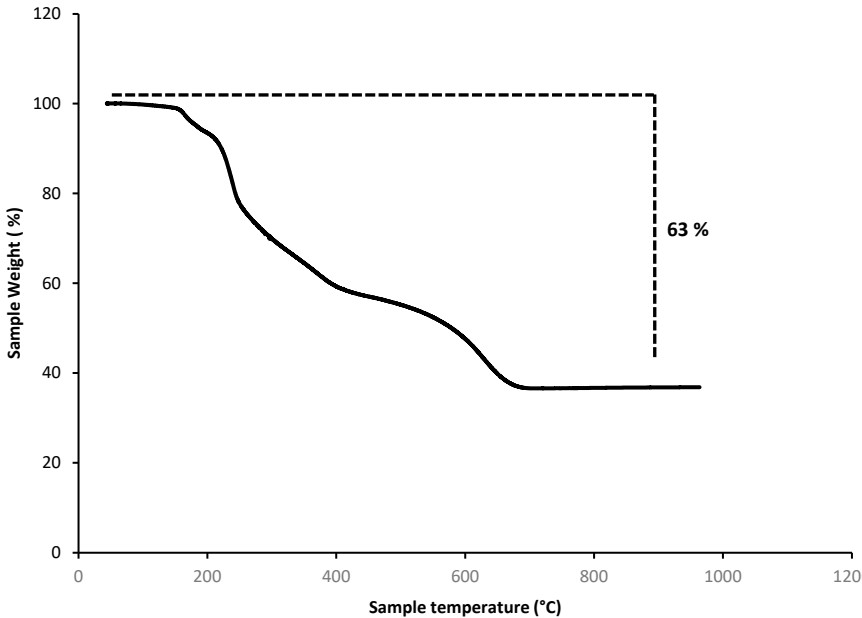

**Figure 3.** Thermogravimetric analysis of a sample of functionalized AuNP powder.

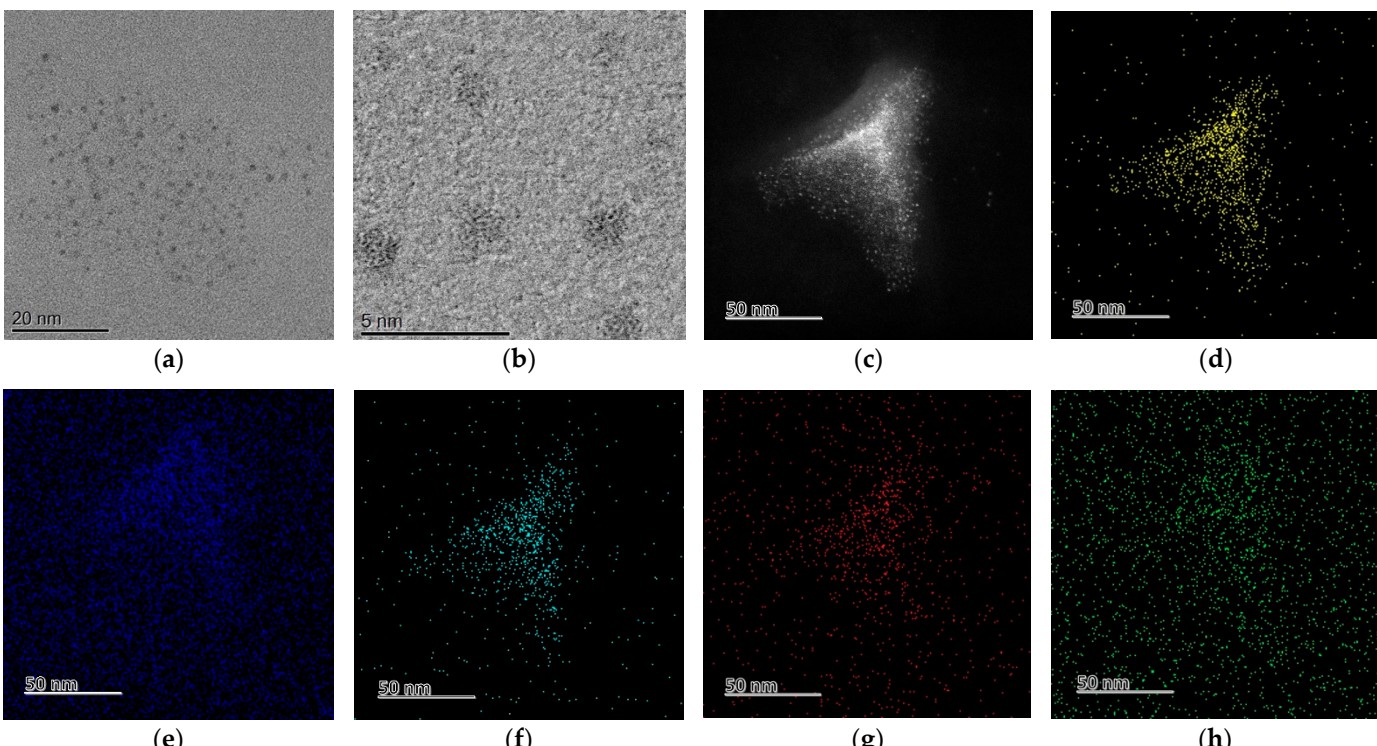

**Figure 4.** Scanning Transmission Electron Microscopy (STEM) images showing Au clusters within nanoparticles (**a**,**b**), and Energy Dispersive Spectroscopy (EDS) maps of the nanoparticle material showing the (**c**) unfiltered signal and the filtered signals from; (**d**) gold; (**e**) carbon; (**f**) sulfur; (**g**) nitrogen; (**h**) oxygen.

### 3.4. Fourier-Transform Infra-Red Spectroscopy (FTIR)

3.4.1. Bonding the Au Cores to a Ligand Coating

IR analysis performed on the Au-ligand bonds show the lack of a strong absorption band at 3500 cm$^{-1}$ and the presence of a moderate absorption band at 3350 cm$^{-1}$ indicates that primary amine groups in L-cysteine have successfully formed secondary amine groups in a ligand structure. There is minimal absorption around 2660 cm$^{-1}$ for the propargylamine-grafted spectrum, suggesting a disulfide bridge from the thiol-capped compound, as expected. Au–S bonds produce absorption bands outside the instrument range. However both samples show absorption bands at 1420 cm$^{-1}$, which is associated with methylene scissoring adjacent to an Au–S bond [29]. These results suggest successful ligand synthesis and bonding to the Au cores with an average size of around 1.5 nm.

3.4.2. Grafting the Propargylamine Crosslinker to Ligand Coating

The IR spectrum of the grafted propargylamine sample is shown in black (Figure 5) and was taken to determine whether propargylamine is successfully grafted to the ligands. A successful reaction was supported by two main criteria: the presence of a terminal alkyne moiety and the absence of any primary amine.

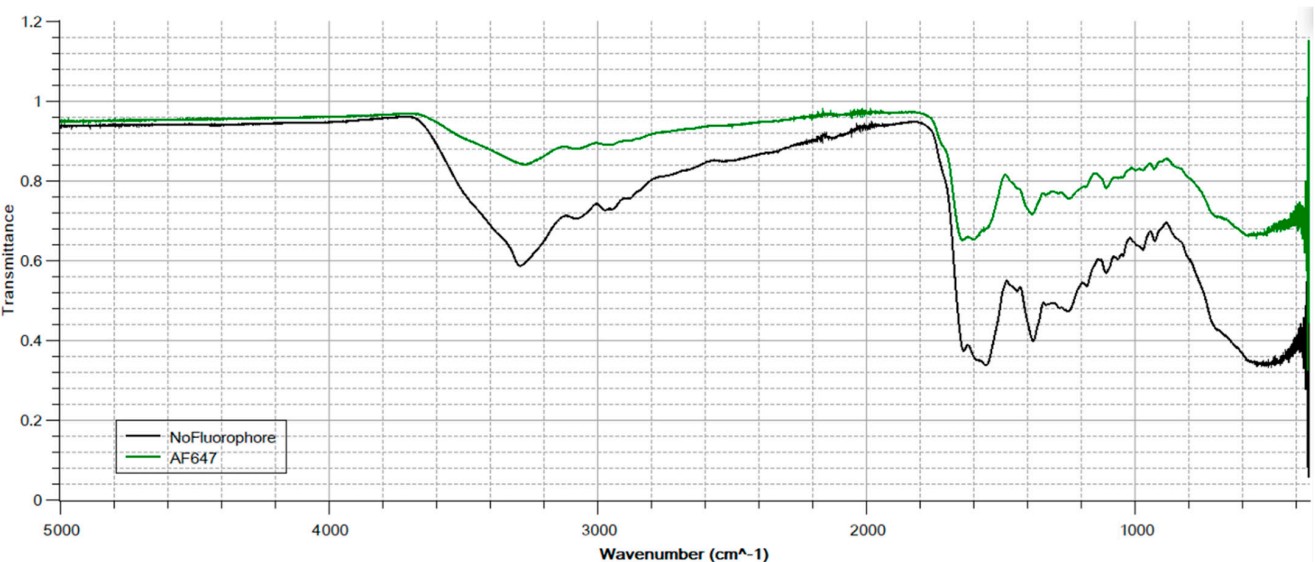

**Figure 5.** Fourier-Transform Infra-Red Spectroscopy graph of AuNP samples with and without the cross-linked fluorophore.

The absence of any primary amine groups indicates that the carboxyl group has undergone a condensation reaction to bind to form a secondary amide. The spectrum indicates a broad absorption band around 3370 cm$^{-1}$ which is likely attributed to strong O–H stretching in alcohols and C–H in an alkyne (Aldrich) rather than any primary amine groups in propargylamine that have not reacted (around 3480 cm$^{-1}$).

At 1635 cm$^{-1}$, both samples suggest C=O stretching of a secondary amide [30], indicating that the propargylamine has grafted successfully. Absorption peaks are present in both samples as this region overlaps with C=C alkene stretching and N-H bending. The contribution from the AF647 compound is indicated by a significant increase in absorption.

The terminal alkyne is seen at 3285 cm$^{-1}$, which correlates to –C≡CH stretching [31–34]. This is a prominent mode in the propargylamine sample, and although this region overlaps with other characteristic bonds, there is a unique and distinct peak at this wavenumber.

A lack of a strong absorption band in the region 2145–2120 cm$^{-1}$ indicates that the C=N double bonds of carbodiimide have successfully broken to facilitate the subsequent amide grafting [35]. The region shows a minor absorption at 2120 cm$^{-1}$ in the propargylamine sample. However, this sits at the lower bound of the carbodiimide region and

corresponds exactly to the wavenumber of a terminal alkyne, where a weak band is expected. As such, minimal absorption in this region can be considered indicative of a successful synthesis procedure.

### 3.4.3. Click Reaction Bonding Crosslinker to AF647 Fluorophore

IR analysis for the fluorophore sample is shown in green and was used to determine whether AF647 has been successfully bonded to the ligands via a copper-catalyzed azide-alkyne cycloaddition (CuAAC) reaction. A successful click chemistry reaction would be indicated by a lack of terminal alkyne bonds and an absence of double-bonded nitrogen in an azide moiety, where the click reaction produces a triazole ring.

The increased absorption for the propargylamine sample at 1435 $cm^{-1}$ is due to the scissoring of the C–H bond in terminal alkynes, as expected. The triazole ring is indicated by an increased absorption around 1720 $cm^{-1}$ [31,33,36].

The fluorophore sample shows a clear decrease in the strong absorption band at 3285 $cm^{-1}$ and at 1440 $cm^{-1}$, which is characteristic of the stretching of a terminal alkyne [31,32,34,37]. An internal alkyne bond would produce a peak between 2260–2190 $cm^{-1}$. This region is devoid of any observable absorption bands, indicating that the carbon triple bond did not remain due to some transformation into a disubstituted alkyne.

The data provided indicates that the reaction mechanism successfully produced a ligand chain which likely has a thiolate moiety where it has fused to the Au core. The change in the IR spectra suggests that the propargylamine grafted to the ligand's carboxyl groups and the fluorophore was bound to the surface via a copper-catalyzed azide-alkyne cycloaddition 'click chemistry' reaction.

### 3.5. Absorption and Emission Spectra of Gold Suspension

Using the SpectraMax Plus 384 Microplate Reader (Molecular Devices, USA), the absorption and emission spectra of the AuNP suspension were probed (Figure 6). The peak absorption was detected at 647 nm, and the peak emission at 673 nm (when excited with 647 nm, negligible scattering). This agrees well with the expected absorption and emission of AF647. The absorbance of the light, however, remains at least at a relative 0.3. This is likely due to the absorption by the AuNPs themselves, which will have a peak in optical density in the range of 550 to 600 nm depending on their size [38].

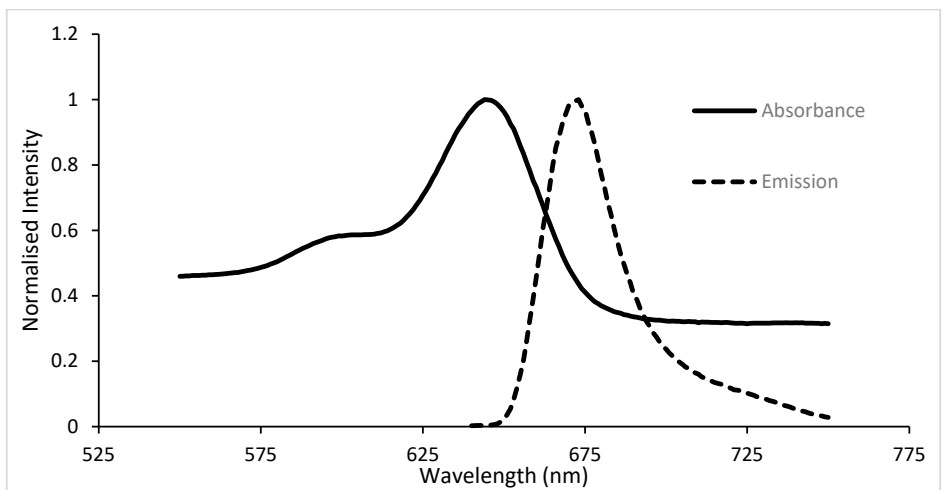

**Figure 6.** Absorbance and emission (relative to the maximum detected output) spectra of the AuNP suspension.

### 3.6. Confocal Imaging to Detect the Nanoparticles within 9LGS

Initial imaging of cells exposed to the gold solution using confocal microscopy in Figure 7 shows an uptake around the cells and the nanoparticles fluorescing. It is important

to note that the excitation of the laser used was 633 nm and, thus, not the ideal excitation wavelength for Alexa Fluor 647. From Figure 7, the nanoparticles can be found surrounding cells and being internalized. Two concentrations were used, 50 μg/mL and 100 μg/mL, to qualitatively compare the relative uptake by cells. As is clear, there is significantly more fluorescence occurring in the sample that contained 100 μg/mL of the AuNP solution. Furthermore, there is no aggregation of the nanoparticles at either concentration. They are well distributed.

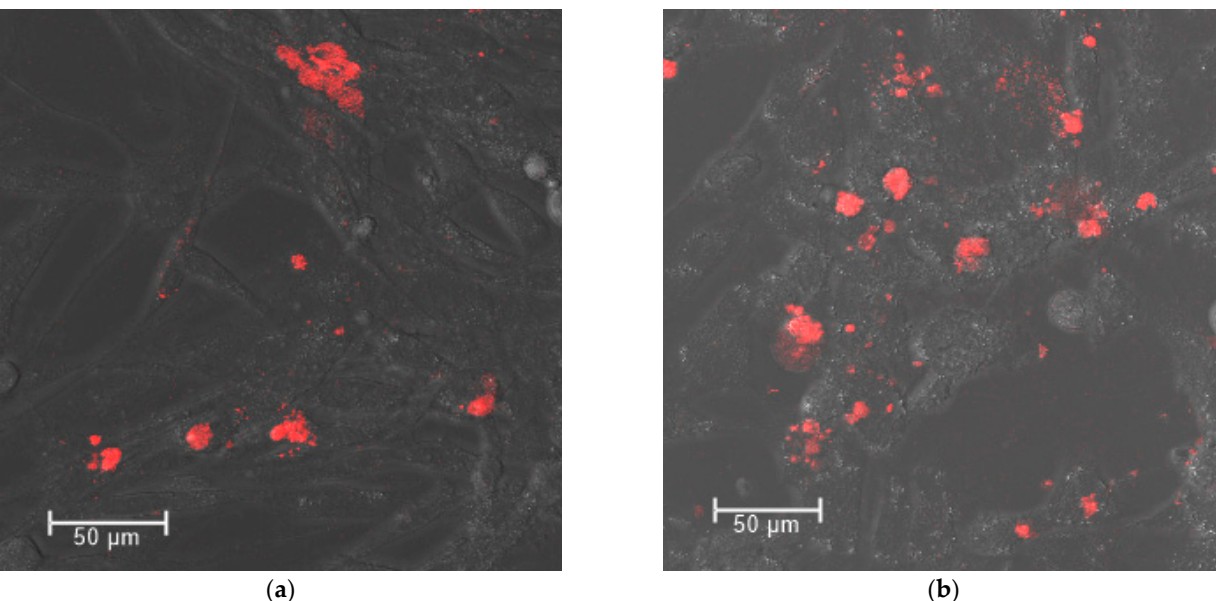

(**a**)                              (**b**)

**Figure 7.** Confocal images of 9LGS cells at 63× magnification exposed to either (**a**) 50 or (**b**) 100 ug/mL of the gold suspension for 24 h. The red areas are the detection of the AF647 fluorophore attached to the nanoparticle against the bright field.

An aliquot of cells exposed to the fluorescent nanoparticles was removed prior to flow cytometry (results in Section 3.7.) to visually confirm the fluorescence inside the cells (Figure 8). Using confocal microscopy with a 633 nm laser, the fluorescence was strictly observed to be within the non-adherent cells. Thus, the nanoparticles are well internalized by the cells and remain as such after washing.

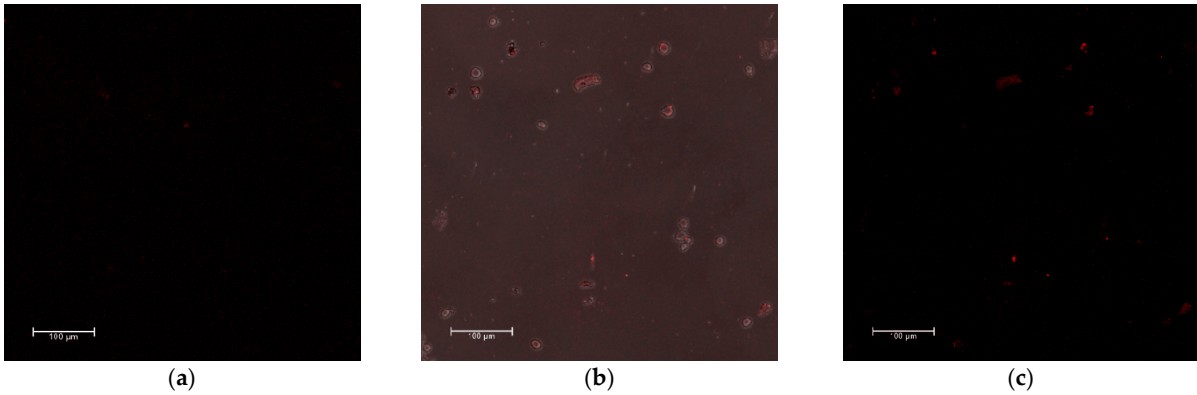

(**a**)                    (**b**)                    (**c**)

**Figure 8.** Detection of AF647 using a 633 nm excitation laser with confocal microscopy at 20× magnification for (**a**) control 9LGS cell population and cells exposed to 50ug/mL gold suspension for 24 h; (**b**). with bright field; (**c**) without bright field. Cells were washed and suspended in PBS (hence they are not adherent). The red areas of AF647 fluorescence are clearly located inside the cells.

### 3.7. Flow Cytometry to Detect the Nanoparticle Uptake and Fluorescence within 9LGS

Flow cytometry supports the findings from confocal imaging, verifying that the nanoparticles do not aggregate in the cellular environment. The side scatters of the cells exposed to 50 µg/mL of the solution for 24 h did not increase compared to the control (Figure 9). The mean side scatters for control cells was $1359 \pm 640$, which is very similar to the side scatter for cells exposed to the gold suspension, $1120 \pm 578$. The non-fluorescent addition, Dihydrodichlorouorescein diacetate (H2DCFDA), will fluoresce when it is hydrolyzed by esterases and oxidized to dichlorofluorescein (DCF) by intracellular ROS. There was no increase in the DCF signal for the cell population exposed to the AuNPs compared to the control (data not shown).

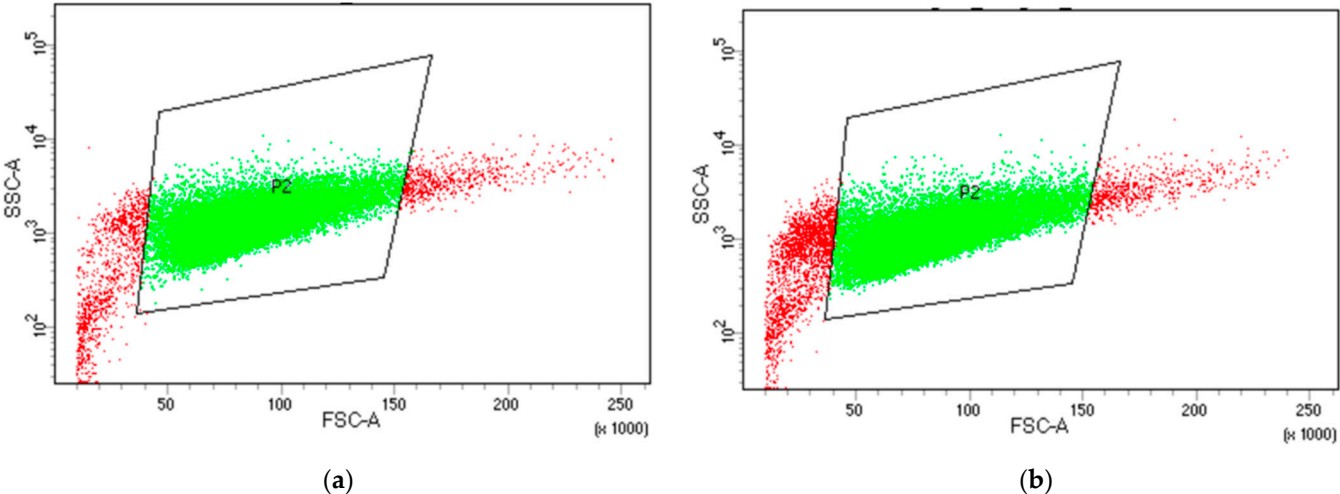

(**a**)　　　　　　　　　　　　　　　　　　　　　　　　　　　　(**b**)

**Figure 9.** Forward scatter vs. side scatter for (**a**) control cell population and (**b**) cells exposed to 50 µg/mL AuNP suspension for 24 h. Forward vs. side scatter is commonly used to identify the size of cells and particulates. If nanoparticles were large or aggregated and internalized, the side scatters would increase dramatically.

Furthermore, the AF647 fluorophore attached to the AuNPs was detectable in cells using a 640 nm laser (Figure 10). The control cells showed almost no fluorescence, but there was significant fluorescence in the cells exposed to the gold suspension. The mean value for control cells was 928, whilst the mean value for cells exposed to the AuNP solution was 16,377 (arbitrary intensity units).

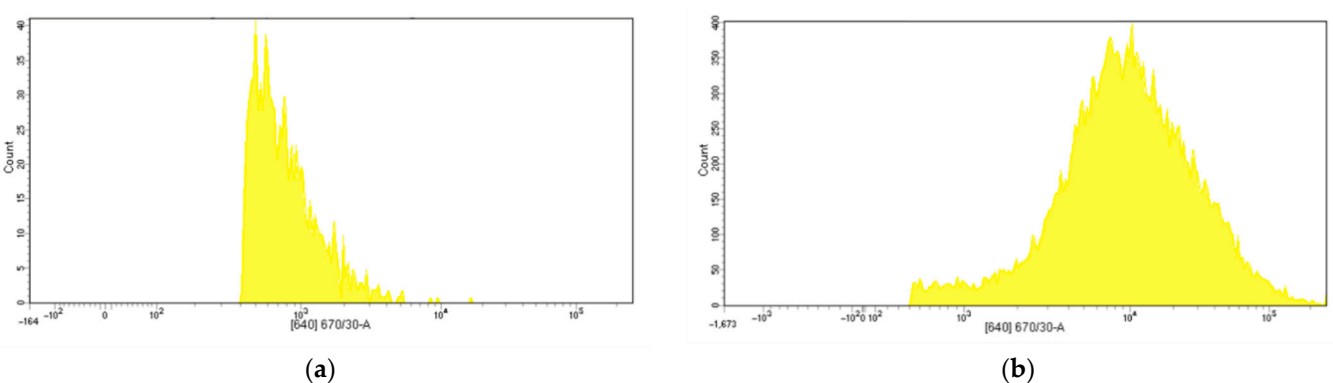

(**a**)　　　　　　　　　　　　　　　　　　　　　　　　　　　　(**b**)

**Figure 10.** Detection of AF647 using a 640 nm excitation laser through flow cytometry for (**a**) control cell population and (**b**) cells exposed to 50 µg/mL gold suspension for 24 h. Note the different scales on each graph for the *y*-axis.

### 3.8. Gold Suspension Cytotoxicity and Orthovoltage Irradiation Survival

The long-term effects of the AuNP suspension were determined using a clonogenic assay. Cytotoxicity experiments show that there is some toxic effect on 9L cells, with survival decreasing to 76 ± 5%. However, this is likely a result of the decreased media volume for cells in the initial 24 h of incubation (the final concentration of the AuNP colloid dispersed in water was 144 µg/mL, such that at 50 µg/mL in the cellular environment, over one-third of the media is replaced, leading to cell starvation). This was proven using live cell imaging, where media was replaced 24 h after the initial incubation with the nanoparticle, and the cells grew to normal confluence.

Clonogenic assay was also used to determine the enhancement of 125 kVp radiation, shown in Figure 11. Error bars represent the standard deviation taken over three repeats, with two triplicates per repeat, and nanoparticle-treated samples were normalized to the un-irradiated nanoparticle control. A pooled *t*-test was performed on the data showing that the surviving fraction for control and AuNP treated cells are significantly different for both 2 and 5 *Gy* irradiation (2 *Gy* $p$ value = $8.83 \times 10^{-5}$ and 5 *Gy* $p$ value = $3.38 \times 10^{-8}$. As is clear, the AuNPs lead to a reduced surviving fraction when compared to the control cells. The radiation enhancement ratio (*RER*) was [39] calculated for each dose.

$$RER_{xGy} = \frac{SF_{xGy,control}}{SF_{xGy,NP}} \tag{1}$$

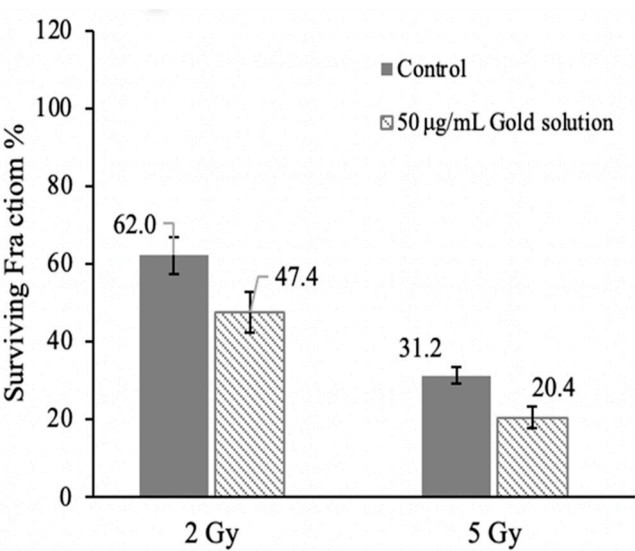

**Figure 11.** Surviving fraction of 9LGS cells untreated and treated with 50 µg/mL of AuNP suspension, irradiated at 2 and 5 Gy 125 kVp compared to control cells. Treatment populations were normalized to their respective unirradiated treated data.

$RER_{2Gy}$ was found to be 1.308 and $RER_{5Gy}$ 1.529. This effect is in line with the enhancement reported in other work utilizing AuNPs in kilovoltage treatments [40].

### 3.9. Double Strand Break Quantification in Irradiated 9LGS with AuNP Suspension

The main mechanism of radiation enhancement by nanoparticles is the induction of double-strand breaks in DNA. This is crucial as breaks that are very close together are difficult for a cell to repair and can lead to apoptosis. To determine their production, γ-H2AX was used to visually locate double-strand breaks immediately following irradiation (20 min) in both treated and untreated cells.

From Figure 12, there is a significant increase in the strength of the signal and number of foci within the cells that were treated with both 5 Gy of 125 kVp irradiation and 50 µg/mL of AuNP solution compared to 5Gy irradiation alone. Using ImageJ, the number of foci

was counted per cell, and the average foci enhancement in treated cells was (1.3 ± 0.1). This enhancement suggests that damage and subsequent cell death are induced very soon after radiation treatment, but as described in Section 3.8, this damage will also continue during the long-term period of the clonogenic assay.

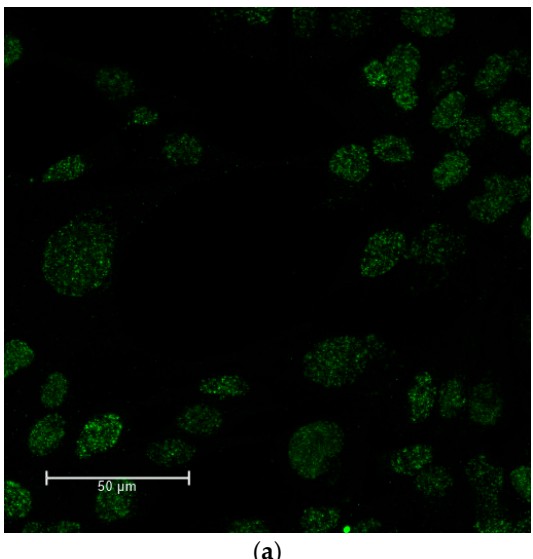 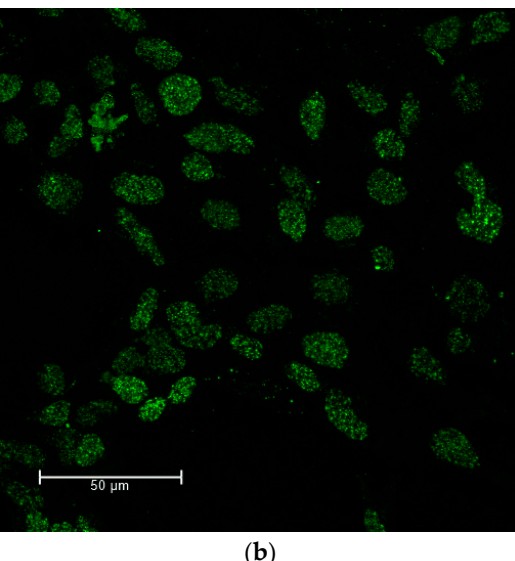

(**a**)                                     (**b**)

**Figure 12.** $\gamma$-H2AX characterization with green representing double-strand break foci. Images are a stack of 50 individual slices at 63× magnification taken from the first instance of foci to the last instance, across the depth of the cells (**a**) 9LGS control and (**b**) with 50 μg/mL of AuNP solution, both irradiated at 5 Gy with 125 kVp.

*3.10. Post Mortem Injection of AuNP Suspension*

A 50 μg/mL concentration of the AuNP suspension that was diluted in DPBS was injected into the flank of a Fischer 344 rat cadaver at an approximate depth of 5 mm. Using optical imaging targeting the AF647 excitation with the IVIS Spectrum CT unit, the fluorescence is obvious around the injected site (Figure 13).

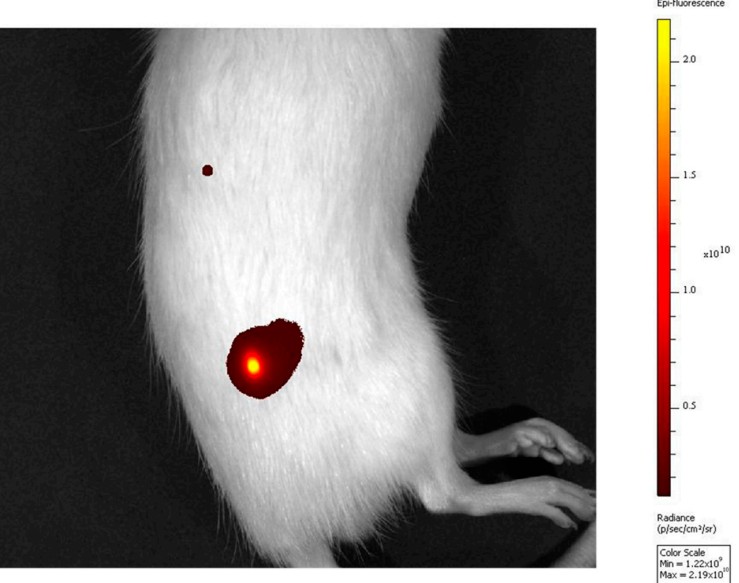

**Figure 13.** Epifluorescence of the AuNP solution was detected surrounding the injection sight in the right flank of a Fischer 344 rat cadaver.

## 4. Discussion

This research endeavored to produce a new fluorescent nanoprobe that is relatively simple to formulate, retains its properties within the cellular environment, and improves image-guided radiation treatment. This type of nanoparticle in suspension is highly desirable in the treatment of cancer, especially for radio-resistant glioblastoma, as it allows for real-time location of the nanoparticle using fluorescence in conjunction, or instead of, computed tomography and delivers significant radiation enhancement.

We have developed a novel process to fabricate fluorescent AuNPs in suspension in four steps. First, L-Cys-DTPA ligand synthesis allows the connection between the gold and the carbon chain by establishing a sulfur bridge. This synthesized ligand L-Cys-DTPA was used to functionalize the gold cations. The metallic core was then built by the reduction in the gold's cations using sodium borohydride. The EDC/NHS coupling reaction forms amide bonds between carboxyl groups of L-Cys-DTPA and propargylamine. Finally, the terminal alkyne group of the ligand allows the fixation of AF647 fluorescent dye with a copper-catalyzed Huisgen 1,3-dipolar cycloaddition of alkynes groups of the ligand to azides groups of the dye. All the synthesis steps were designed to produce a strong anchorage of chemical functions of interest. This type of synthesis is, therefore, very flexible and achieves very stable surface functionalization of nanoparticles and is an alternative to other similar methods [41,42] while remaining within the specifications of the biological application in terms of low intrinsic toxicity, but also in terms of the fluorescence necessary for the diagnostic functions of the suspension.

We were able to prove the success of our process through Transmission Electron Microscopy and IR spectroscopy. STEM results suggest the successful synthesis of gold-core particles as high-mass aggregates supported by a matrix of organic elements. FTIR spectroscopy elucidates the structure of the ligand coating and confirms the Cu-catalyzed Azide-Alkyne Cycloaddition of the AF647 fluorescent dye through a cross-linked molecule. The data provided indicate that the reaction mechanism successfully grafted propargylamine to the ligand's carboxyl groups. This was evidenced by a lack of primary amide groups and broken terminal alkyne bonds to form a triazole ring.

We further demonstrated the efficacy of the nanoparticle suspension as a radio enhancer. The AuNP suspension was proven to be stable and was readily internalized by 9LGS and did not aggregate, even over a 24 h period. The nanoparticles were simple to locate in vitro due to the AF647 fluorescent dye that remained attached to the nanoparticle itself. This fluorescence could also be detected using confocal imaging and flow cytometry by targeting their fluorescence, even after the cells had been thoroughly washed. When cells exposed to the AuNP solution were irradiated using 125 kVp, the radiation was enhanced at both 2 and 5 Gy doses resulting in a decrease in overall cell survival, correlating to a 35% increase in cell death at 5 Gy kilovoltage, in line with other radiation studies with AuNPs [43–46]. The presence of the nanoparticles also produced an increase in short-term damage, as evidenced by the clear increase in double-strand breaks only 20 min after irradiation (an increase factor of $1.3 \pm 0.1$).

Finally, the nanoparticles were visible when injected at muscular depth into a rodent cadaver and imaged using optical imaging. From these images, there is a significant hot spot at the injection site, at a depth of approximately 5 mm. This acts as a proof of concept for this suspension as a diagnostic imaging solution.

## 5. Conclusions

Click-chemistry using a copper-catalyzed azide-alkyne reaction is a simple synthesis to produce a fluorescent labeled AuNP in suspension. This suspension is stable in the cellular environment, and the attached fluorophore allows for a simple location of the nanoparticle. The nanoparticle behaves as expected as a radio enhancer at orthovoltage energies. Future work will investigate the pharmacokinetics and tumor-targeted imaging power of the suspension in live animals. The concentration of the nanoparticle suspension must be a consideration as it resulted in some cytotoxic effects. As a result, the ratio of

media to suspension within the cellular environment. This will be addressed in the future production of this theranostic tool and will make it more effective and safer for use in vivo.

**Supplementary Materials:** The following supporting information can be downloaded at: https://www.mdpi.com/article/10.3390/jnt4010003/s1, Figure S1: Elemental mapping of AuNP suspension.

**Author Contributions:** Conceptualization, M.T., M.L., S.C. and K.O.A.; methodology, S.V., C.M., L.S. and A.O. formal analysis, S.V., C.M. and A.O.; investigation, S.V., A.O., L.S., M.V., E.E., C.H. and A.K.; data curation, S.V., A.O., L.S. and C.M.; writing—original draft preparation, S.V. and A.O.; writing—review and editing, M.T., K.O.A., S.C., C.M. and M.V.; visualization, S.V. and A.O.; project administration, M.T. and K.O.A., funding acquisition, M.T. All authors have read and agreed to the published version of the manuscript.

**Funding:** The authors acknowledge the generous donation from a donor facilitated by Shaye Hiscocks of Advancement Operation at the University of Wollongong. They also acknowledge the financial support of the Australian Government Research Training Program scholarship (S.V.).

**Institutional Review Board Statement:** The animal study protocol was approved by the ethics committee at the University of Wollongong, (approval number AE20/02, date of approval: 10 March 2020).

**Informed Consent Statement:** Not applicable.

**Data Availability Statement:** The data presented in this study are available on request from the corresponding author. The data are not publicly available due to ethical reasons.

**Acknowledgments:** The authors acknowledge the time, access to, and technical and scientific assistance from the Prince of Wales Hospital, Sydney, the Illawarra Health and Medical Research Institute, Wollongong, the School of Chemistry at the University of Wollongong (UOW), and the Fluorescence Analysis Facility and Animal House in Molecular Horizons at UOW.

**Conflicts of Interest:** The authors declare no conflict of interest.

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
