# Peer review of "Fluorescent Gold Nanoparticles in Suspension as an Efficient Theranostic Agent for Highly Radio-Resistant Cancer Cells"

_jnt, doi:10.3390/jnt4010003_

Round 1

Reviewer 1 Report

It was a study about the fabrication and evaluation of the theranostic effects of fluorescent Au NPs on radio-resistant cancer cells. Here are some comments on this study that should be considered before publication:

1.      Please mention used materials as a separate subsection of part 2.

2.      Please add references related to the synthesis methods.

3.      Please rewrite the following sentence:

“An aqueous solution of N-(3-dimethylaminopropyl)-N’-ethylcarbodiimide (EDC) 128 and N-hydroxysuccinimide (NHS) (1 to 2 in molar ratio) was added to a colloidal suspen- 129 sion of gold nanoparticles functionalized at 1g/L.”

4.      Please mention the brand and manufacturer country of all used instruments.

5.      Please write the stability test process in the method section.

6.      Why pH doesn’t affect the aggregation of the nanoparticles.

7.      TGA results need to better description. Normally water loss occurs under 100 °C. please check and improve this section.

8.      Please delete figure 4-a and mix Figures 4 and 5. Please add the mean size of fabricated particles.

9.      Please mention the place of amid bond in the FTIR results in section 3.4.2.

10.  Why did you use 50 µg/ml of the NPs for the flow cytometry test?

11.  Please explain more about the intracellular ROS production test in the method.

12.  “However, this is likely a result of the decreased media volume for cells in the initial 24 hours of incubation (the final concentration of the gold nanoparticle solution was 144 µg/mL, such that at 50 µg/mL in the cellular environment, over one third of the volume is NPs rather than media).” I think this is not a good explanation. Please find another reason for that. Were the NP dispersed in the solution or in colloid form?

13.  Please test the cytotoxicity of the combination use of NPs and radiation.

14.  Please compare the results of your study with other similar ones in the discussion section.

Reviewer 2 Report

The authors have described the GNP synthesis method and its possible utilization of cancer theranostics. The experiments are poorly designed and without statistical analysis. There is a significant change required in the manuscript;

1. Method section is poorly written;

a.       There is no information regarding the source/Name of companies from where all the chemicals were purchased.

b.       No information on the concentration of chemicals used in the Synthesis of [Au(L-Cys-DTPA)] nanoparticles. How much gold salt or NaBh4 was used? What was the concentration of propargylamine? There are various occasions where the methods are poorly written. The authors need to provide all the information so that the work can be reproduced.

2.  The authors should use a primary cell line as a control to compare the cytotoxicity. The comparison of cytotoxicity between tumor and primary cells should be discussed.

3. There is no mention of Figure 1 in the manuscript.

4.  It would be better to include pictures of GNP solutions at various pH along with Figure 2 for clarity. Furthermore, it is unclear why the UV-vis spectra have no peak for colloidal GNP. In general, colloidal GNP should have a peculiar peak near 500~550nm.

5. The authors should include a graph of elemental mapping.

6. All confocal images should include a scale bar. There is a significant fluorescence in Fig. 9; the background should be adjusted.

7. There is no information regarding the magnification of confocal images.

8. How were 50 µg/mL and 100 µg/ml 385 concentrations selected for the uptake studies?

9. Are the results in Figure 12 significant? The authors should include the statistical significance.

10.   How many animals were used in the experiment? There is no information on the statistical significance of the animal study. The authors should include more animals in the images including control. Further, animals were not shaved, so it will be interesting to see autofluorescence.

11.   There is no information on the statistical analysis of data in the entire manuscript. How many replicates were used in the study? What kind of statistical analysis was performed?

12.   The authors should include the ethical committee approval information in the manuscript for animal testing.

Reviewer 3 Report

Dear authors,

Thank you for the interesting work. I think it fits the scope of JNT.

I have some comments and questions regarding the manuscript.

 1. In the Introduction, the authors review some applications of different kinds of nanoparticles. From my point of view, it is necessary to give also some references on possible destructive effect of nanoparticles towards biomolecules, for example:

doi: 10.1117/1.JBO.20.4.047004,

doi: 10.3103/S0027134914060058.

Please, enlarge the introduction.

 2. In the Introduction, it is unclear whether the authors discuss nanoparticles from gold or from other materials (lines 53, 77, 81, 83, 88, 90, 96, etc.).

 3. Abbreviation AuNPs for gold nanoparticles is introduced in line 50. Please, use this abbreviation in subsequent text (lines 93, 94, 101, 107, 125, 130, 171, 172, 177, 205, etc.)

 4. Please, show the scale in Fig.5.

 5. Please, show the error bars in Figs. 2, 3, 6, and 7.

 6. The suspension of nanoparticles is insufficiently characterized.

What is the exact physical composition of the suspension?

What are the concentrations of the homogeneous parts of the suspension?

What is the mean size of the nanoparticles in suspension?

What is the dispersion of nanoparticle sizes?

What is the shape of nanoparticle in suspension?

 7. Point 2.3. Please, specify the ATR crystal in FTIR experiments.

 8. Point 3.4. Is the interaction of nanoparticle sample with the surface of ATR crystal was taken into account? The concentrations of components can be changed during spectra accumulation. It may cause (together with surface interactions between particles and crystal) absorption changes seen in Fig. 6. With this respect, it seems to be very useful to give a couple of references on the works where the possible interaction of the sample with ATR crystal is studied, for example:

doi: 10.3103/S002713491806022X.

 9. Point 3.5. How did the authors take into account scattering in the sample? Obviously, transmittance of the sample depends on both absorption and scattering coefficients. Fig. 7 shows specifically “absorption”. Please, discuss it in the text.

 10. Please, indicate at what excitation wavelength emission spectrum in Fig.7 was measured.

 11. Fig.7. What are the units on Y-axes? Probably, that is absorbance, not absorption?

 12. Section “Conclusions” is absent. Please, summarize your results.

Round 2

Reviewer 1 Report

Some of the comments are not addressed which are mentioned as follows:

1-     The list of materials used in the paper, their purchased company, and manufacturer country should be written as a separate sub-heading in section 2.

2-     Please mention the manufacturer country for FTIR instrument.

3-     Still there are some grammatical mistakes in the text.

4-     Why doesn’t pH affect the aggregation of the nanoparticles?

5-      The justification of the authors for TGA results is not acceptable yet. Does the platform have a porous structure?

6-     Please compare the results of your study with other similar ones in the discussion section.” Where did you add these data?

Reviewer 2 Report

The authors have answered all the questions raised during the peer-review process.

Author Response

Many thanks for reviewing our manuscript.

Reviewer 3 Report

Dear authors,

thank you for the answers. I think that the manuscript looks much better now.

 I would like to clear the point concerning error bars on the figures. Your answer is as follows.

“No error bars available because all the characterization have been only one time because for TGA, the sample is destroyed. Similar for other instances. This is not possible for figures 6 and 7 (similar to this article doi.org/10.1002/cnma.202100359).”

You are absolutely wright that quite a lot of papers do not show error bars. Except for only some of them, this position is pseudo-scientific. In most cases, one certain number in physics is meaningless. For example, I can state that free fall acceleration is equal to 30 m/s2. But if give an accuracy of +– 25 m/s2, this number will be absolutely correct. In the case of biological samples, correct experiments are very complicated. Normal variations of biological samples can be very impressive. Thus, one spectrum do not provide adequate information.

Anyway, you promise in your answer to continue the work. So, I hope you will take into account my point of view.

Author Response

Thank you for reviewing our manuscript. We will take on your feedback and suggestions for future work and submissions.